# Effect of Welding Parameters on Friction Stir Welded Ti–6Al–4V Joints: Temperature, Microstructure and Mechanical Properties

**Junping Li, Fujun Cao and Yifu Shen ***

College of Materials Science and Technology, Nanjing University of Aeronautics and Astronautics, 29 Jiangjun Road, Nanjing 211106, China; lijunping@nuaa.edu.cn (J.L.); CFJ0510@163.com (F.C.)
* Correspondence: yfshen_nuaa@hotmail.com; Tel.: +86-025-8489-5940

**Abstract:** In this study, friction stir welding (FSW) of 2-mm-thick Ti–6Al–4V alloy plates was performed using a newly designed friction tool—and the effect of rotation speed and welding speed on microstructure and mechanical properties of the joints were investigated. A simulation model for FSW temperature field calculation was developed, and the effect of rotation speed and welding speed on the temperature field was investigated by experimental and numeric methods. The results show that the rotation speed has a dominant effect on peak temperature, while welding speed determines the dwell time of the weld exposed to high temperatures. In addition, the influence of process parameters on the microstructure of the joints was investigated using optical and scanning electron microscopy. The results revealed that there was a phase transformation in the stir zone during welding. The final microstructure of the stir zone was fully lamellar ($\alpha + \beta$) structure, and the heat affection zone had a bimodal microstructure consisting of prior equiaxed $\alpha$ and lamellar ($\alpha + \beta$) structure. Both rotation speed and welding speed affect the grain size of the weld. Lower peak temperature with decreasing spindle speed and/or shorter dwell time with increasing feed rate could produce finer grains in the stir zone of the joints, thereby could lead to higher microhardness value and the tensile strength of the joints.

**Keywords:** friction stir welding; Ti–6Al–4V; temperature; microstructure; mechanical property

## 1. Introduction

Friction stir welding (FSW) was developed as a solid–state joining technology in 1991 by The Welding Institute (TWI) [1]. This welding technique is implemented using a non-consumable friction tool that is plunged into the materials at a certain rotation speed and then moves forward with a given feed rate, as shown in Figure 1. The tool is moving at a higher rotation speed to obtain sufficient friction heat to soften the material in the weld zone. Then the softened material is squeezed, stirred and mixed under the tool, and then the materials bonded together at the welding zone finally [2]. So, it is a solid state joining technology, but metallurgical bonding of materials is realized ultimately. Nowadays, due to its many special advantages, FSW has become a common joining technique for Al, Mg and other light metals [3]. However, for high strength and high melting temperature metals, such as Ti, Ni and steel, higher requirements are put forward for FSW technology [4]. For example, higher hardness and high-temperature wear resistance tool materials, special FSW equipment with cooling system, and so on. To overcome these limits, people have done some researches. Some researchers have FSWed pure titanium and titanium alloy using pure tungsten and tungsten alloys [5,6], sintered TiC [7] and pcBN [8] as welding tool materials. Zhou et al. [9] and Fall et al. [10] have developed some equipment with water cooling system for FSW of titanium alloys.

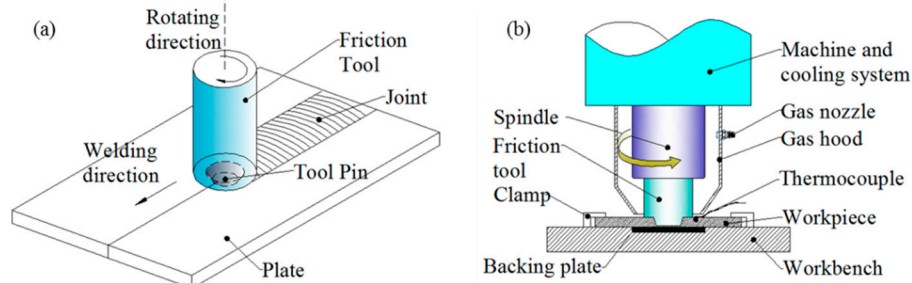

**Figure 1.** (**a**) Schematic illustration of friction stir welding (FSW), (**b**) schematic diagram of FSW equipment of Ti alloys.

Titanium and its alloys have long been widely used in the aerospace, chemical and medical fields, due to its superior strength to weight ratio, good corrosion resistance and excellent biocompatibility. With the wide use of titanium alloys, the welding of Ti and its alloys is becoming increasing important. So, researchers have payed increasing attention to the FSW of titanium alloys, due to that it could avoid some problems in fusion welding methods, such as brittle cast structure, large distortion and residual stress [2,3,11]. Ti–6Al–4V alloy, as the most widely used material in titanium alloy family due to its unique performances, researchers have done many FSW studies on this material, mainly focused on the welding parameters, microstructures and mechanical properties of joints. For example, Edwards et al. [12] had identified welding parameters for FSW Ti–6Al–4V butt joints with 3 to 12 mm in thickness. It was reported that the primary parameters determining welding quality were rotation speed and welding speed. Zhou et al. [9] have investigated the effect of tool rotation speed on the microstructure and mechanical properties of FSWed Ti–6Al–4V alloy joints. The results show that the rotation speed had a significant effect on the microstructure and mechanical properties of the Ti–6Al–4V joints. Sungook et al. [13] have studied the effect of rotation speed on the microstructure and the texture evolution of FSW Ti–6Al–4V joints and confirmed that the microstructure in the stir zone exhibited inhomogeneous distribution. All these studies have yielded some important knowledge on FSW of Ti–6Al–4V alloys. However, there are some problem to be addressed and overcome before the industrial application of FSW to titanium alloys. The chief problem is the relationship between welding parameters, temperature field, microstructure and mechanical properties, which is needing a systematic study. Hence, this study is focused on this problem, to evaluate the range of the welding parameters for butt welding of Ti–6Al–4V alloys thin plates, and to build the relationship between welding parameters, temperature field, microstructure and mechanical properties for the FSW Ti–6Al–4V butt joints.

## 2. Materials and Methods

In this study, 2-mm-thick Ti–6Al–4V plates with a dimension of 200 mm × 200 mm were used as experimental materials. The as-received Ti–6Al–4V alloy plates have a typical mill-annealed microstructure, characterized by the elongated α grain with β-phase at boundaries, as shown in Figure 2a,b. The α-phase is colored red as base structure and β-phase is illustrated with blue color in the phase-map, as shown in Figure 2b. It is clear that the β-phase is mainly at the grain boundaries. The average grain size of the microstructure is 8.7 um, as presented in Figure 2b. Moreover, the Ti–6Al–4V alloy has the following chemical composition (wt%): 5.91—Al, 0.091—Fe, 0.010—C, 3.90—V, 0.15—O, 0.004—H, 0.006—N, balance—Ti.

All the metallographic samples were made and polished according to standard procedures and etched with Keller reagent (4 mL HF, 6 mL HCl, 10 mL $HNO_3$ and 190 mL $H_2O$). Microstructures were observed with an OLYMPUS optical microscope (OM, Olympus-DSX, Olympus, Tokyo, Japan), scanning electron microscope system (SEM, FEI Quanta430, FEI, Portland, OR, USA) and electron backscattering diffraction (EBSD, FEI, Portland, OR, USA). Microhardness was measured with a Vickers hardness tester (HXS-1000AY, Laizhou Huayin Testing Instrument Co., Ltd., Laizhou, China), under

500-N load and 15-s dwell time. The tensile tests were performed on the YLS-900 testing machine (Suzhou Yuehai Corp., Suzhou, China) with a crosshead speed of 1 mm/min at room temperature.

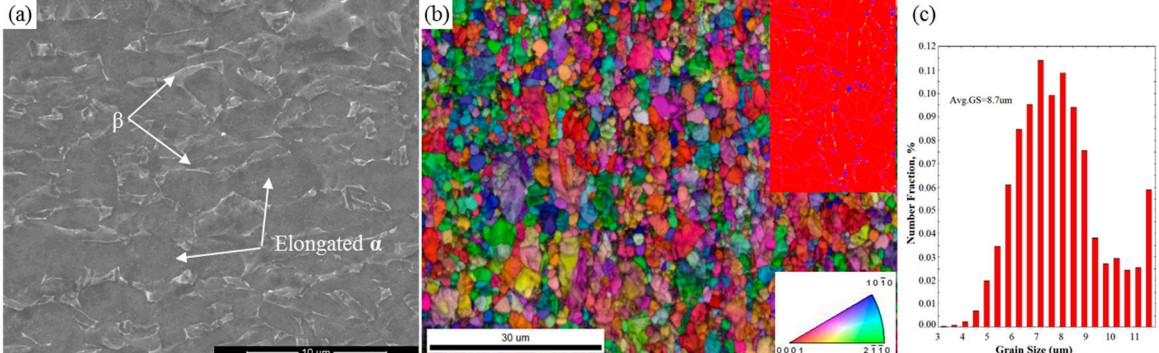

**Figure 2.** As-received Ti–6Al–4V alloy: (**a**) SEM image of microstructure, (**b**) electron backscattering diffraction (EBSD) image of inverse pole figure (IPF) map and phase map (with respect to TD), (**c**) distribution of grain size acquired using the EBSD.

The FSW equipment (Nanjing Gaoteng Corp., Nanjing, China) in this work is a special numeric control welding machine, with some sensing device to measure the real-time displacement during welding. Argon gas (≥99.9%) was used as the shielding gas to prevent contamination of the joint in the welding processes, as illustrated in Figure 1b. Because titanium alloys are easy to react with $O_2$ and $N_2$ in the air at above 500 °C to produce brittle compounds, such as $TiO_2$, $TiN$, which could deteriorate the properties of the joints. The flow rate of the argon gas is 2 L $min^{-1}$. Considering the economic problem of industrial application and high performance requirements, the welding tool was designed as a combined welding tool in this work, as shown in the Figure 3. Tungsten base alloy (W-25% Re) was selected as the tool pin material because of its excellent high-temperature wear resistance. A nickel-based super-alloy (GH4043) was selected as the material of tool shoulder, because of its good high-temperature impact resistance. In addition, these two materials do not react with titanium alloys at high temperatures. In order to generate enough heat to weld high melting point titanium alloy, a larger diameter friction tool was designed in this study. Based on the previous researches [14–16], the tool shoulder diameter was designed as 26 mm, and the length of the tool pin was 1.9 mm with a taper from 16 to 14 mm, as shown in Figure 3. Thus, the ratio of shoulder-to-pin diameter is 1.55, which is smaller than that of the conventional tools: 3. The tool diameters distribution of conventional tool was illustrated in Figure 4 [2,4,8,17–20]. Additionally, due to its bigger diameters, the tool will produce a more stable temperature field and will have a higher impact strength.

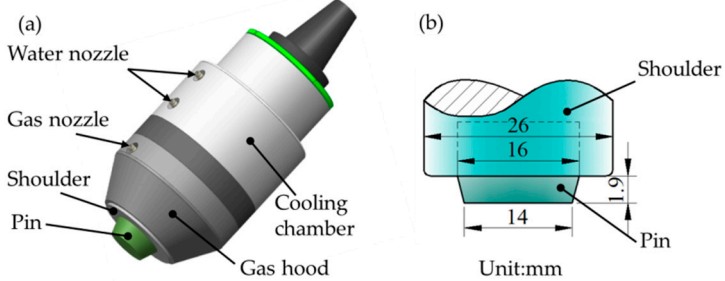

**Figure 3.** Schematic of FSW tool. (**a**) Schematic for friction tool structure, (**b**) dimensions of the friction head.

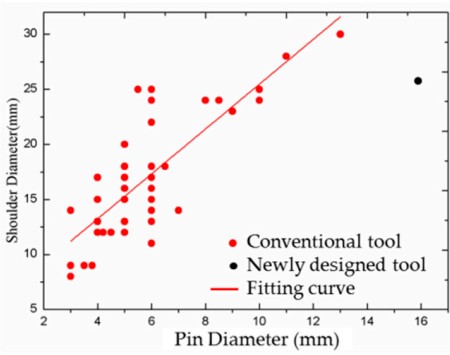

**Figure 4.** Comparison of the tool diameters distribution.

As mentioned earlier, FSW technology is a multiparameter weld process. According to the previous studies [14,21–24], welding speed and rotational speed are the key parameters that affect the welding quality of FSWed Ti–6Al–4V joints. Hence, in this work, to investigate the effect of the welding parameters on the welding quality, the rotation speed and welding speed are set as the variable, as listed in Table 1. Moreover, other parameters are set to constants as follows: tool tilt angle 0°, axial force 30 KN, plunging depth 0.1 mm and plunging speed 0.02 mm/s.

**Table 1.** Welding parameters matrix and corresponding macro-quality of joint.

| Rotational Speed (rpm) | Welding Speed (mm/min) | | | |
|---|---|---|---|---|
| | 10 | 20 | 40 | 60 |
| 700 | Defect free | With defect | - | - |
| 900 | Defect free | Defect free | Defect free | With defect |
| 1100 | Defect free | Defect free | Defect free | With defect |
| 1300 | With defect | With defect | - | - |

The welding temperature at the center of the weld zone of FSW is difficult to monitor in real time. To study the relationship between welding parameters and the temperature field about FSW of Ti–6Al–4V in detail, a thermal numeric model and its detailed calculating method of the heat generation were proposed in this work. In this model, the friction heat between the friction tool and work-piece is defined as the heat source, so the friction heat generation is from three different regions, as illustrated in Figure 5.

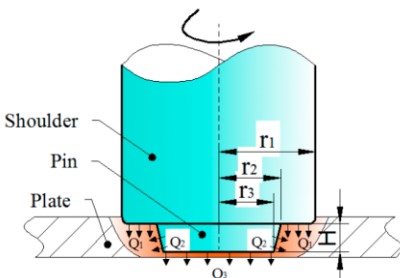

**Figure 5.** Schematic diagram of heat source distribution of friction tool.

Where $Q_1$ is the heat generated under the tool shoulder, $Q_2$ is the heat generated at the tool pin side and $Q_3$ is the heat generated from the tool pin tip. Hence the total heat generation is given by Equation (1). In addition, the tool in this work has a flat shoulder surface, tapered cylindrical pin side surface and flat pin tip surface. Hence, the calculating model is a modified version based on previous literature [25–27].

$$Q_{total} = Q_1 + Q_2 + Q_3 \tag{1}$$

Some assumptions must be made for this thermal numeric model, as below:

1.  The workpiece material is isotropic and homogenous, and no melting occurs during the welding.
2.  The heat generation from material plastic deformation is negligible. The heat transfer between the workpiece and clamp tools is also negligible.
3.  The thermal boundary conditions are symmetrical along the weld center-line.

In order to calculate the heat generation of $Q_1$, $Q_2$, and $Q_3$, the calculated model is given by the Equations (2)–(4).

$$Q_1 = 4\pi^2 \cdot N \cdot \mu \cdot F \cdot \left(r_1^3 - r_2^3\right)/3 \tag{2}$$

$$Q_2 = \pi^2 \cdot N \cdot \mu \cdot F \cdot r_2^2 \cdot h/15 \tag{3}$$

$$Q_3 = 4\pi^2 \cdot N \cdot \mu \cdot F \cdot r_3^3/3 \tag{4}$$

Thus, the total heat generation expression simplifies to Equation (5) ($r_2$ and $r_3$ are approximately equal):

$$Q_{total} = \pi^2 \cdot N \cdot \mu \cdot F \cdot \left(20r_1^3 + h \cdot r_2^2\right)/15 \tag{5}$$

where $N$ is the rotation speed of the tool and $\mu$ is the friction coefficient. $F$ is the axial force of the tool, set as 30 KN. Moreover, $r_1$, $r_2$, $r_3$ and $h$ are tool dimensions, as shown in Figure 5. The coefficient of friction ($\mu$) varies with temperature. However, in this model, it was set as 0.5, as an approximate value. In this research, the MSC. Marc software (MSC Software, LA, USA) was utilized to simulate the temperature field characteristics.

## 3. Results

### 3.1. Temperature

In the FSW process, according to some reports [28–30], the welding parameters directly affect the temperature characteristics of the welding zone. Moreover, in turn, the welding temperature controls the microstructure type, grain size and mechanical properties of the joints. Therefore, it is necessary to explore the relationship between the welding parameter and temperature characteristics.

Figure 6a is the thermal cycles of FSW of Ti–6Al–4V at different rotation speeds and at a welding speed of 20 mm/min. To verify the accuracy of the thermal simulation model, the actual temperature histories of a group of FSW experiment was measured by thermocouples. The temperatures of actual tested results and simulated results are presented as curves 1 and 2 in Figure 6a, both at the welding parameters of 900 rpm and 20 mm/min. The actual test point and its corresponding simulated point are located 1 mm away from the weld edge. Although curve 1 has a certain time delay compared with curve 2, peak temperatures of the two curves are similar. Hence, the simulated temperature is basically consistent with the actual measured temperature, which indicated that the developed calculation model for the FSW temperature field is dependable in this work. In addition, from Figure 6a, it is clear that the peak temperature increases with the increase of the rotation speed. Moreover, the peak temperature is approximately proportional to the rotation speed, as indicated in Figure 6b. This is due to the increase in friction heat generation per unit area with the increase of the rotation speed. As shown in Figure 6a, when the rotation speed was up to 900 rpm at the welding speed of 20 mm/min, the peak temperature could reach 1000 °C, which exceeds the β-transit temperature (995 °C). This means that the welding zone material will undergo phase transformation when the rotational speed is above 900 rpm.

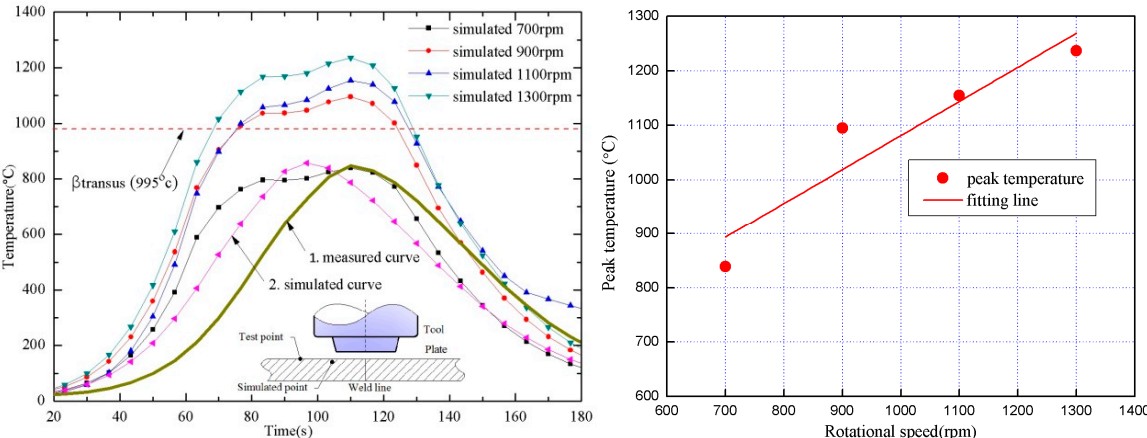

**Figure 6.** Temperature profiles (at welding speed of 20 mm/min). (**a**) Temperature history profiles at various rotational speed; (**b**) peak temperature distribution at different rotational speeds.

Figure 7 is the thermal cycles of FSW of Ti–6Al–4V at a constant rotation speed and various welding speeds. It shows that when the welding speed increases, the peak temperature slightly decreases. Thus, the peak temperature is inversely proportional to the welding speed, which is due to the reduction of friction time per unit area of the weld zone when the welding speed increases. In addition, Figure 7 also shows that when the welding speed increases, the dwell time of high temperature (above β-transit temperature 995 °C) decreases sharply. It means that the welding speed significantly affected the exposure time of the welding zone to the high temperature, which is due to the shortening of the friction time per unit length when the welding speed increase. In addition, from Figures 6 and 7, the fluctuation range of peak temperature from various welding speeds at the constant rotation speed is smaller than that from various rotation speeds. Hence, it can be concluded that the welding speed has a minimal influence on the peak temperature of the welding.

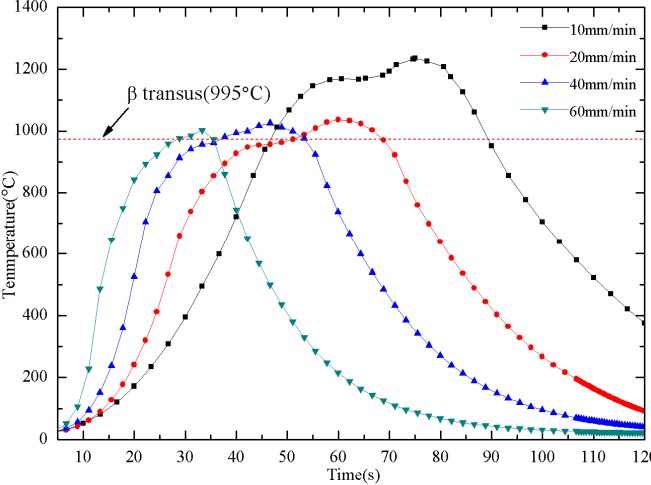

**Figure 7.** Temperature history profiles at various welding speed (all at rotational speed of 1100 rpm).

### 3.2. Weld Morphology and Microstructure

The macro and micromorphologies of the weld are criteria for evaluating whether the welding process is reasonable or not. Figure 8 shows the typical weld surface appearance and cross-section morphology of FSWed Ti–6Al–4V joints. As shown in Figure 8a,b, there are no obvious defects on the surface and cross-section of the weld produced at 1100 rpm and 20 mm/min. However, at 700 rpm and 60 mm/min, some tunnel and unconnected defects appear on the weld surface, as shown in Figure 8c,d. These two types of defects are the main defects in the defective samples of this experiment. These defects

were due to insufficient or excess heat input in the welding, because of unreasonable process parameters. The same defects also appeared in the FSWed Ti–6Al–4V joints of some studies [6,29].

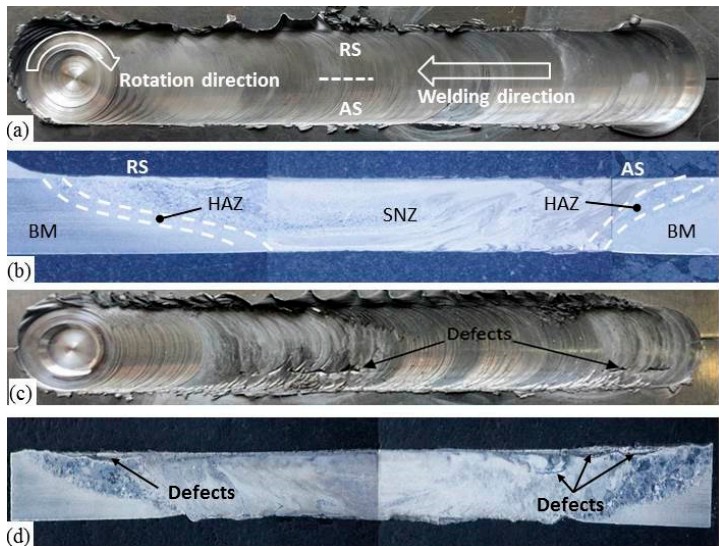

**Figure 8.** Joint surface morphology (**a**) at 1100 rpm, 20 mm/min; (**c**) at 700 rpm, 20 mm/min; Macroscopic view of joints cross-section (**b**) at 1100 rpm, 20 mm/min; (**d**) at 700 rpm, 20 mm/min.

In addition, the macroscopic view of joints cross-section presents a "bowl" shape, which is due to the different degrees of plastic deformation and material flow in various regions of the section. As illustrated in Figure 8b, the cross-section morphology could be divided into three areas: stir zone (SZ), heat affected zone (HAZ) and base material (BM). Nevertheless, some researchers reported that it also has a thermal–mechanical affected zone (TMAZ) in addition to the former three regions [12,13]. Because TMAZ is the very narrow transition zone between the SZ and HAZ, and it is difficult to distinguish the difference of the microstructure between HAZ and TMAZ. So, in this research, TMAZ was also treated as HAZ. Moreover, as shown in Figure 8b, there is an clear boundary between the HAZ and the SZ on the advancing side (AS), but there is an unclear boundary on the retreating side (RS). This is due to the different deformation and flow behavior of the welding materials created by the tool shoulder on both sides. The same phenomenon has been reported by Edwards [30]. In addition, it is noticed from Figure 4b,d that there are no dark spots, white spots or contaminated regions on the weld cross-section, which means that there is no obvious wear debris from the friction tool. Hence, combined with weld quality in Table 1, it could draw a conclusion that the defect-free FSWed Ti–6Al–4V joints could be obtained by appropriate welding parameters with the new friction tool in this work.

The optical micrographs of HAZ, taken from the joints produced at different welding parameters, are given in Figure 9. It is clear that with the increase of rotation speed and welding speed, there are some differences in the microstructure of the HAZ. As shown in Figure 9a, the microstructures of the HAZ is the equiaxed microstructure, which is similar to that of the BM, as indicated in Figure 2. However, as the rotation speed increasing, lamellar microstructure appears in the HAZ in Figure 9b,c. It means that there was a microstructure transformation in the HAZ. This indicates that the peak temperatures in the SZ exceeded the β-phase transformation temperature in the welding and phase transformation (β-phase→α-phase) occurred during the cooling stage. Similar results have also been confirmed by A. Fall [10] and R.S Mishra [31]. However, the grain size of all the HAZ in Figure 9b,d–f are similar. Hence, the effect of welding speed on the grain size of HAZ is not obvious, which is due to the short dwell time of HAZ at high temperature and small deformation of materials of the HAZ. Hence, the grains cannot fully grow in HAZ during welding.

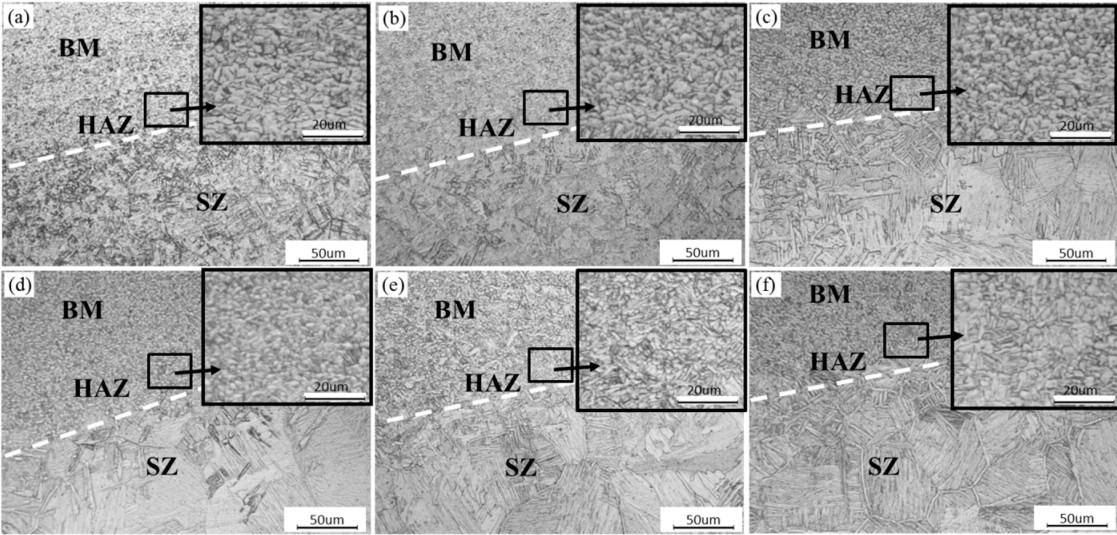

**Figure 9.** Microstructure of HAZ. (**a**) At 700 rpm—10 mm/min; (**b**) at 900 rpm—10 mm/min; (**c**) at 1100 rpm—10 mm/min; (**d**) at 900 rpm—20 mm/min; (**e**) at 900 rpm—40 mm/min; (**f**) at 900 rpm—60 mm/min.

Figure 10 shows the optical micrographs of the SZ, taken from the center of the joints produced at various rotation speeds and welding speeds. Figure 10a is the microstructure of SZ produced at 700 rpm-10 mm/min, which is a heterogeneous microstructure with fragmented α-phase and thin β-phase boundary and the initial elongated microstructure of BM disappears. Figure 10b,c are the microstructure of SZ produced at 900 rpm and 1100 rpm, which is a typical Widmanstätten structure with fully lamellar (α + β) grain. This lamellar microstructure is composed of α phase grain boundary and lamellar (α + β) colonies. Therefore, it means that the materials in SZ have experienced the phase transition (β-phase→α + β) in the welding at 900 rpm and 1100 rpm. So, this result also confirms that the welding temperature in SZ exceeds the β-transit temperature when the rotation speed was up to 900 rpm, which is in good agreement with the former temperature simulation results in Figure 6. The same research results also have been confirmed by other studies [32,33]. So, it could be concluded that the rotation speed not only determines the peak temperatures in the welding, but also control the microstructure type of SZ. Figure 10d–f show the variation in the microstructure of the SZ with an increase in welding speed. It appears that when the welding speed increase, not only the grain size of the prior β-phase decreases gradually, but also the lamellar microstructure gets finer. The microstructure of SZ is fully transformed into lamellar α + β colonies at 900 rpm and 20 mm/min. However, when the welding speed increased to 60 mm/min at 900 rpm, not only the grain size of prior β-phase reduced, but also the fraction of precipitated α-phase in the β-regions decreased. This is because that faster tool moving speed would result in shorter dwell time at high temperature and reducing the heat input, thus decrease grain size of prior β-phase and creating lower fraction α-phase lamellar microstructure during β→α + β transformation in the end. Hence, welding speed has a significant effect on the grain size of the SZ, but it has little influence on the microstructure type of SZ.

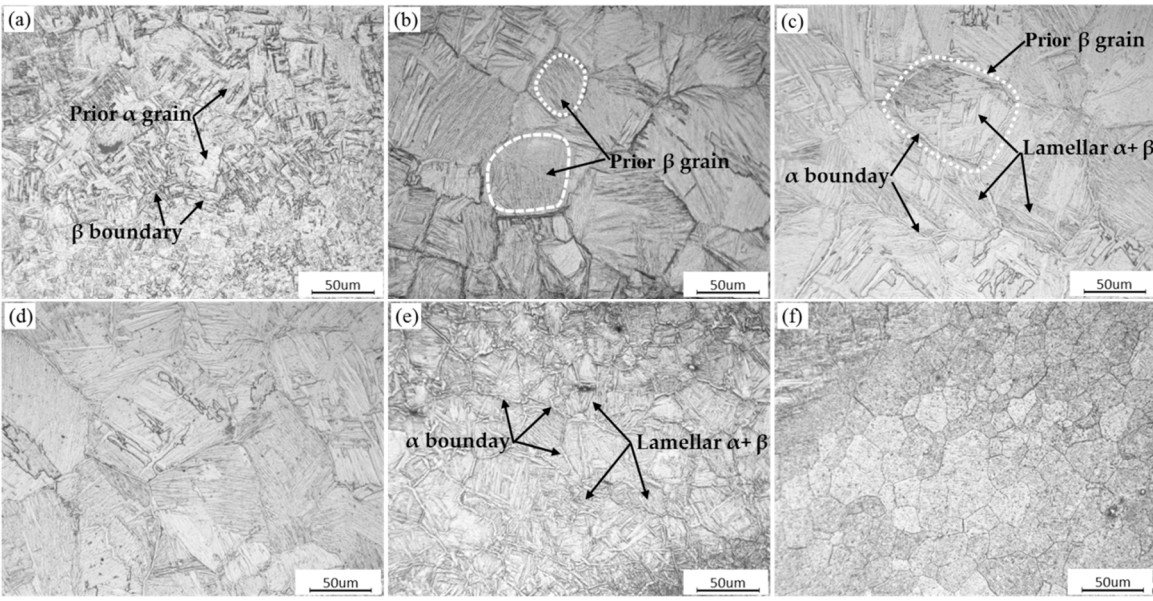

**Figure 10.** Micrographs of SZ. (**a**) At 700 rpm—10 mm/min, (**b**) at 900 rpm—10 mm/min, (**c**) at 1100 rpm—10 mm/min, (**d**) at 900 rpm—20 mm/min, (**e**) at 900 rpm—40 mm/min, (**f**) at 900 rpm—60 mm/min.

EBSD maps of as received Ti–6Al–4V alloy and the joints at various welding parameters are presented in Figure 11. The microstructure of as received Ti–6Al–4V alloy was elongated grains along the rolling direction, and the average grain size was about 8.5 um, as shown in Figure 11a,b. Figure 11c is the grain boundary map of HAZ at 900 rpm and 10 mm/min. It shows that the grain boundaries become not smooth and a small amount of lamellar structure appears on the initial structure. Moreover, Figure 11d presents more lamellar structure and some distorted structure. As the rotation speed increase to 1100 rpm, the SZ is characterized by a fully lamellar microstructure in Figure 11f. So, the comparison of microstructure between BM (Figure 11a) and SZ (Figure 11d,f) indicates that there was a phase transformation in the SZ during welding. The occurrence of phase transformation during FSW of Ti–6Al–4V alloy has also been reported by Zhang et al. [34] and Yoon S. et al. [13]. Moreover, the final lamellar structure of SZ is composed of layers of needle-shaped acicular α-phase and retained β, as shown in Figure 11d,f. In addition, by comparison of Figure 11d,f, it can be found that with the increase of rotational speed, the phase transformation in the SZ becomes more sufficient, and the lamellar structures get coarser. The same research results have also been reported in other literature [13,29,35]. Figure 11c appears a bimodal microstructure consisting of prior equiaxed α and many lamellar (α + β) grains, which means that the materials in HAZ performed insufficient phase transformation at 900 rpm. Because there was not enough time and heat input for the growth of β grains. Figure 11e appears more lamellar grains than that in Figure 11c and the bimodal microstructure of HAZ characterized by mixed basket-weave and colony morphologies. Since, the welding temperature increases as the rotational speed increases, which could promote the phase transformation in the HAZ. However, due to the high cooling rate of thin plates, the phase transformation is still insufficient in HAZ. As a result, the microstructure of HAZ is still a bimodal structure even at a higher rotational speed.

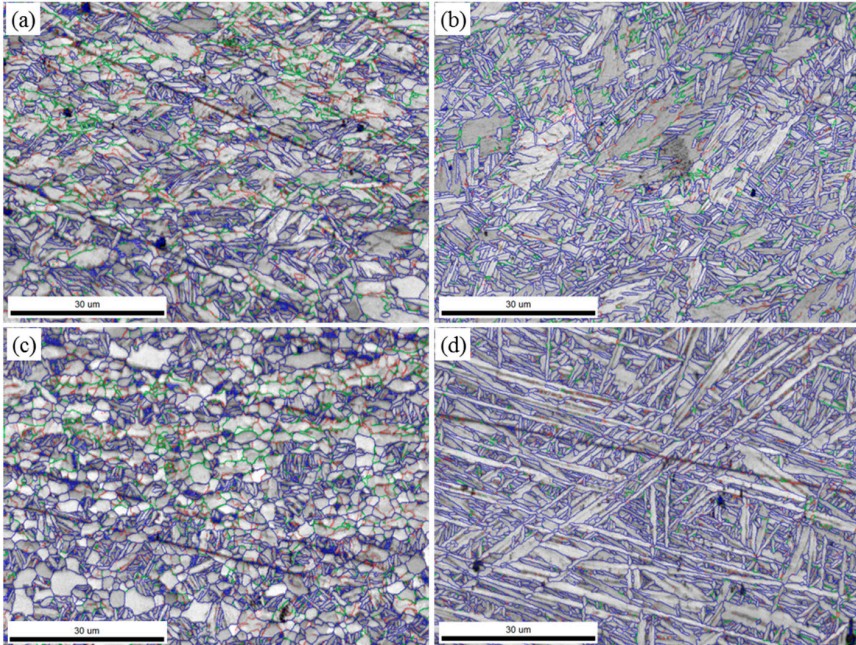

**Figure 11.** EBSD (image quality + grain boundary) maps of: (**a**) HZA, at 900 rpm—10 mm/min (**b**) SZ, at 900 rpm—10 mm/min (**c**) HAZ, at 1100 rpm—10 mm/min (**d**) SZ; at 1100 rpm—10 mm/min.

The grain size distribution of SZ at various welding speeds and rotation speeds are presented in Figure 12. It shows that when the rotation speed is at 700 rpm, the grain size of SZ is similar to BM. However, as the rotation speed increase from 700 to 1100 rpm, the average grain size of the SZ grows significantly from 8 to 24.6 μm. But varying the welding speed from 10 to 60 mm/min at a set rotation speed of 1100 rpm, the grain size of SZ reduced from 24.6 to 11.5 μm. Thus, as the rotation speed increases, the grain size of weld nugget increases. However, the welding speed is inversely proportional to the grain size. So, the welding speed and the rotation speed have a certain influence on the grain size of the SZ. As mentioned above, this is due to the phase transformation of the SZ materials at high temperature during welding. Based on the studies of Zhou Li et al. [36], Edwards et al. [12] and Li et al. [28], the process of phase transformation of Ti–6Al–4V alloy in FSW is as follows: as the friction tool moving forward, the welding material in front of the tool was preheated. Then the temperature of the material in the welding zone exceeded the β-transus temperature (995 °C) when the tool reached the preheated area. As the tool continued moving and stirring, the prior equiaxed (α + β) structure of BM transformed into single β-phase grains and the grains were grown at the same time. In this process, the size of transformed β grains was governed by the temperature and dwell time above the β-transus temperature. Thus, decrease the welding temperature or shortening dwell time could reduce the grain size of transformed β grains. Then in the cooling stage after stirring, the transformed β-phase will transform into α-phase at the inner and grain boundary as an intergranular lath phase. However, the cooling rate is relatively higher because the sheet is very thin. Hence, Gnofam et al. [37] supposed this cooling process as an air quenching of Ti–6Al–4V alloy from the β-phase field. Thus, there is insufficient time for the growth of α grains in the β→α process during the cooling stage. In the end, the transformed-β grins transformed into acicular α grains and retained β grains, and formed α lamellar colony in the end. Hence the final grain size of SZ is largely determined by the size of the prior β grain. In general, decreasing rotation speed or increasing welding speed means reducing the heat input and shortening dwell time at high temperature, which in turn decrease the grain size of SZ. The same conclusion also was reported by Mashinini et al. [29]. In addition, according to the results of other researches [1,38], reducing the degree of deformation will produce a lower driving force for dynamic recrystallization, leading to an increase in the recrystallized grain size based on the general principles of recrystallization. At the same time, the shorter dwell time at high temperatures

not only causes better dynamic recrystallization and smaller recrystallized grain, but also reduces the driving energy of prior β grain growth. Hence, welding parameters have a certain effect on the dynamic recrystallization of the microstructure, which in turn affects the final grain size of SZ.

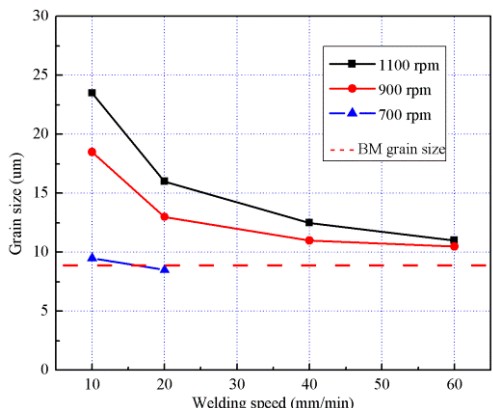

**Figure 12.** Effect of parameters on the average grain size of stir zone (SZ).

Hence, it could be concluded that the heat generation of the FSW process has a strong correlation to the rotational speed and welding speed. The heating-rate and peak temperature are directly controlled by rotation speed, and which in turn dominates the type of microstructure of the weld nugget. Moreover, the welding speed determines the dwell time of the materials exposed at high temperature and thus it directly affects the grain size of SZ. In general, lower rotation speeds and higher welding speeds were expected to produce finer grains structure in the weld, because of the lower peak temperature with decreasing spindle speed and/or shorter dwell time with increasing feed rate. Both of lower peak temperatures and shorter exposure times could produce smaller grain size microstructures. However, the influence of rotation speed on the grain size of SZ is greater than that of welding speed. Since the peak temperature of the welding zone is controlled by the spindle speed, which is the main factor affecting the final grain size of the joints. These results also correlate to the observation of other research [28,30].

### 3.3. Mechanical Property

Figure 13 is the average microhardness variation of the SZ at various rotation speeds and welding speeds. It is clear that all the SZ presented the lower hardness value than that of BM, and the hardness value decreases with the increase of the rotation speed or decreasing the welding speed from 40 to 10 mm/min. According to the Hall–Patch relationship, the reason for this microhardness value difference is due to the difference in the microstructure of the joints at various welding conditions. As shown in Figures 11 and 12, the grains at SZ were coarsened obviously as the rotation speed increasing and/or decreasing welding speed, which is the main reason for the decrease in hardness value of the SZ. In addition, the deformed materials could be softened during recovery or recrystallization in the welding, because of the dislocation density reduction and microstructure coarsening [1,31]. Hence, this is another reason for the decrease in the hardness value of the SZ. Third, according to some research reports [28,37], the acicular α grains are softer than that of prior finer equiaxed α grains. Hence, at a higher rotation speed, a large number of lamellar structures (α + β) formed in the SZ, which is the additional reason for decrease hardness of SZ. However, under the welding speed 60 min/min, respectively, the average hardness of SZ was slightly lower than that using the welding speed of 40 mm/min in Figure 13, It was due to the significant reduction of the α phase fraction in the β-regions, as shown in Figure 10f. As reported by Li et al. [39] that the lower α phase fraction in the β-regions could reduce the hardness of dual α + β phase microstructure. In general, the FSW produced dual α + β phase microstructure with fine β-regions and the lamellar α/β characterization had higher hardness. Thus, the rotation speed and welding speed have a significant effect on the hardness value

of the SZ. Additionally, there is no significant variation in the hardness of HAZ at various conditions because the HAZ is thinner and that has a similar microstructure between each other.

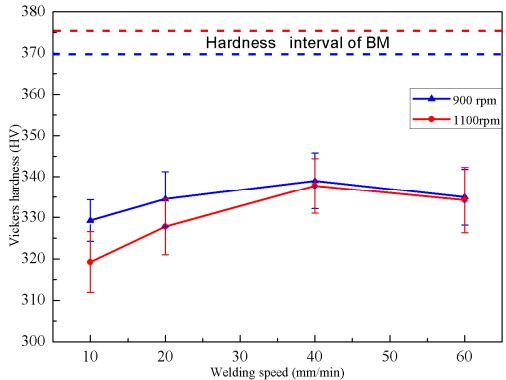

**Figure 13.** Effect of parameters on Vickers-hardness of FSWed Ti–6Al–4V joints.

Figure 14 shows the variation in ultimate tensile strength (UTS) of the joints at various welding conditions. It is clear that all joints have lower tensile strength than that of the parent metal, and the UTS of the joints decreases with the increasing of rotation speed. Thus, the effect of rotation speed on tensile strength is similar to that of the effect of microhardness. To some extent, the strength and hardness of metal materials are related to the final microstructure of materials. Hence, the effect of rotation speed and welding speed on the tensile strength could be explained by the microscopic defects and microstructure coarsening. In addition, the thickness of the weld zone is thinned because of the forming of the weld flash. This is another reason for the UTS of the joint lower than that of the parent material.

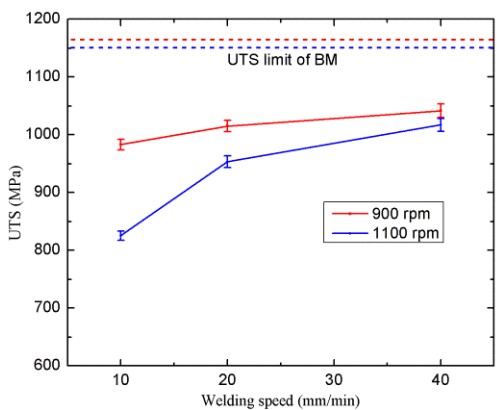

**Figure 14.** Effect of parameters on UTS of FSWed Ti–6Al–4V joints.

From Figures 13 and 14, it also could be observed that the tensile strength and microhardness have similar variation trends relative to process parameters. As the rotation speed increases and/or welding speed decreases, both the tensile strength and microhardness values decrease. It could be concluded that the combined effects of microstructure type, grain size and defect formation obviously determined the mechanical properties of the joints. Thus, well performance of the FSW joints depends on appropriate welding conditions, such as rotation speed, welding speed, tools design and so on.

## 4. Conclusions

In this work, the FSW of 2-mm-thick Ti–6Al–4V sheets was successfully carried out by a new friction tool and the relationship between the welding parameters, temperatures and microstructure was introduced and discussed in detail. The following conclusions could be drawn:

1.　The rotation speed dominates the peak temperature of the welding. The welding speed determines the dwell time of the SZ exposed at the high temperature, but it has little influence on the peak temperature;
2.　When the rotation speed was above 900 rpm, the temperature in the weld nugget could exceed the β-transus temperature (995 °C) and the materials in SZ will perform a phase transition. The final structure of the SZ is a fully lamellar microstructure and the HAZ has a bimodal microstructure characterized by basket-weave and colony morphologies;
3.　Rotation speed and welding speed affect the grain size of the weld nugget. Welding speed is the dominant factor in the grain coarsening. Lower peak temperature with decreasing spindle speed and/or shorter dwell time with increasing welding speed could produce finer grains in the weld;
4.　The tensile strength and microhardness of the joints are closely related to the rotation speed and welding speed. The tensile strength and the microhardness values will decrease as increasing the welding speed and/or reducing the rotational speed.

**Author Contributions:** Conceptualization, J.L., Y.S.; methodology, J.L.; software, J.L.; validation, J.L.; formal analysis, J.L.; investigation, J.L.; resources, J.L.; data curation, J.L., F.C.; writing—original draft preparation, J.L.; writing—review and editing, J.L.; visualization, J.L.; supervision, J.L.; project administration, J.L., Y.S.; funding acquisition, J.L., Y.S. All authors have read and agreed to the published version of the manuscript.

**Funding:** This work was funded by the National Natural Science Foundation of China (Grant No. 51475232).

**Acknowledgments:** This work is also supported by the Priority Academic Program Development of Jiangsu Higher Education Institutions (PAPD).

**Conflicts of Interest:** The authors declare no conflict of interest.

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
