# Peer review of "Effect of Welding Parameters on Friction Stir Welded Ti–6Al–4V Joints: Temperature, Microstructure and Mechanical Properties"

_metals, doi:10.3390/met10070940_

Round 1

Reviewer 1 Report

The quality of the welded joint is assessed not only by mechanical
properties, but also by visual inspection. Fig.8 presented Joint
surface morphology, where the weld is very well below the level
of the base material. The edges of the welded joint are of poor
quality. It is necessary to machine them mechanically. In further
experiments, it is necessary to adjust the mandrel so that no
further treatment of the weld surface is required.

Author Response

Dear professor and editor,

   Thank you for your accurate comments and good suggestions, we have addressed all of the comments and suggestions, as  highlighted section in the revised manuscript. Please see the attachment.

   Thank you very much again.

Reviewer 2 Report

 Overall the study is well executed and presented. Though the contribution has some linguistic shortcomings and offers little findings not yet presented in the referenced works, it offers a valid summary. The Temperature model needs to be reevaluated, as it might fit well for this limited application, but neglecting deformation (and deformation induced heating), varying friction coefficients and interface contact conditions does not produce a proper equivalent energy source.

In chapter 1 l77 ff a concluding summary of the state of the art would be benefical over the repetition of base material properties.

In chapter 2 l100 Fig 1 should be referenced, as well as the effects of oxide layer formation on resulting weld properties and oxid layer formation temperatures for the alloy.

L116 the formatting of the reference need to be adjusted.

The conclusion drawn in chapter 3 l173 disregards the pressure component of the phase balance. The transition temperature is influenced by welding conditions, such as pressure and shear.

Chapter 3 l 214 references the incorrect figure (Fig4 instead of Fig. 8).

It would be beneficial to appy the same color scheme to all images in Fig 11.

Labels on Fig 4, 6, 7, 11b, Table1 need to be adjusted.

Author Response

(The authors gave the same response as above.)

Reviewer 3 Report

The manuscript has studied the use of newly developed FSW tools in the joining of Ti6Al4V plates in different rotation speeds and welding speeds. There are good results and story in this paper however, there are some comments as follows:

1- It’s not emphasized in your introduction what is new in your study? Friction stir welding of Titanium alloys is not a recent study as you mentioned in your introduction “Recently, as a new technique, FSW was tried to be applied for titanium alloys welding, especially to 44 the most widely used Ti-6Al-4V alloy”, your first reference which introduced FSW process is for 2005 and we have papers regarding FSW of titanium alloys in the same year:

  • Lee, Won-Bae, et al. "Microstructural investigation of friction stir welded pure titanium." Materials Letters26 (2005): 3315-3318.

Even it’s a publication two years before that:

  • Ramirez, Antonio J., and Mary C. Juhas. "Microstructural evolution in Ti-6Al-4V friction stir welds." Materials Science Forum. Vol. 426. Trans Tech Publications Ltd., Zurich-Uetikon, Switzerland, 2003.

There is a great review paper in this regard:

  • Gangwar, Kapil, and M. Ramulu. "Friction stir welding of titanium alloys: a review." Materials & Design141 (2018): 230-255.

Therefore, please indicate what is the lack of knowledge and what is the contribution to knowledge in your paper.

2- Describe the designed tool for your FSW system and emphasize its benefits in comparison with previous FSW tools.

3- Is Figure 2.b and 11.a indicate the same sample? Where is the second phase in Figure 11.a? I think it would be better if you could bring a phase map of this sample and show two phases in as-received material.

4- Where is your tool size in Figure 4? Could you show it in this figure to have a better comparison?

5- In figure 9 the evolution of HAZ is not clear. Is it possible to have another figure with higher resolution and magnification and discuss this variation? Also, it would be better visually if you have all BM on one side and all SZ on the other side (it’s not like that now)

6- How many tensile tests were performed for plotting Figure 14? Since the variation is not very much from 20 to 40 mm/min welding speed, the trend is not acceptable. Also, bring the error bar for this figure.

7- in Figure 10-f, It seems that in the prior β grain boundary there is a connected second phase, is that transformed β to α phase or residual β phase? According to figure 7 highest welding speed has the least amount of transformed β, so how could you justify this second phase?

8- What’s the reason for decreasing in hardness value from 40mm/min to 60 mm/min welding speed in Figure 13?

9- I just curious what’s the average grain size at 60mm/min for 900 and 1100 rpm welding conditions in Fig. 12? Could you add it to this figure and discuss the reasons.

10- How could you interpret the different variations in tensile strength and microhardness between figures 13 and 14 from 20mm/min to 40mm/min conditions? The increase in microhardness does not affect tensile strength. Explain this by microstructures.

Author Response

(The authors gave the same response as above.)

Round 2

Reviewer 3 Report

Thanks for your answers. In this current form, your manuscript is good to publish.